# 🦅 Hawk: Leveraging Spatial Context for Faster Autoregressive Text-to-Image Generation

**Zhi-Kai Chen**[1,2]     **Jun-Peng Jiang**[1,2]     **Han-Jia Ye**[1,2]     **De-Chuan Zhan**[1,2]*

[1] School of Artificial Intelligence, Nanjing University, China
[2] National Key Laboratory for Novel Software Technology, Nanjing University, China
{chenzk, jiangjp, yehj, zhandc}@lamda.nju.edu.cn

## Abstract

Autoregressive (AR) image generation models are capable of producing high-fidelity images but often suffer from slow inference due to their inherently sequential, token-by-token decoding process. Speculative decoding, which employs a lightweight draft model to approximate the output of a larger AR model, has shown promise in accelerating text generation without compromising quality. However, its application to image generation remains largely underexplored. The challenges stem from a significantly larger sampling space, which complicates the alignment between the draft and target model outputs, coupled with the inadequate use of the two-dimensional spatial structure inherent in images, thereby limiting the modeling of local dependencies. To overcome these challenges, we introduce Hawk, a new approach that harnesses the spatial structure of images to guide the speculative model toward more accurate and efficient predictions. Experimental results on multiple text-to-image benchmarks demonstrate a 1.71× speedup over standard AR models, while preserving both image fidelity and diversity.

## 1 Introduction

The field of image generation has witnessed significant progress [37, 20, 45], especially with the emergence of autoregressive (AR) models that demonstrate strong capabilities in generating high-quality images [48]. These models have been applied in various domains, including super-resolution [13], image completion [43], style transfer [7], image segmentation [28], and image editing [6].

Despite the impressive performance of autoregressive (AR) models in generating high-fidelity images, their slow inference speed remains a significant obstacle for real-world deployment. This limitation primarily stems from their inherently sequential nature—generating images token by token—which leads to low throughput and high latency. While existing approaches, such as non-autoregressive generation methods [14, 20, 38] and model compression techniques like pruning [44, 25] or quantization [2, 50], can accelerate inference, they often do so at the cost of degraded image quality. This raises the question: is it possible to speed up inference without compromising generation fidelity?

Speculative decoding has already demonstrated its effectiveness in accelerating autoregressive text generation models without compromising output quality [19, 33, 12, 10]. This technique employs a two-model framework: a lightweight draft model first generates a sequence of tokens, which are then partially or fully verified by a larger, high-quality target model. If the target model confirms the draft's predictions, multiple tokens can be accepted in a single step, thus reducing the number of sequential forward passes and accelerating inference. Given its success in the text domain, it is natural to consider applying speculative decoding to AR image generation as a means to achieve faster inference without sacrificing image quality.

---

*Corresponding author, email: zhandc@lamda.nju.edu.cn.

39th Conference on Neural Information Processing Systems (NeurIPS 2025).

Although AR-based text and image generation share many underlying principles, key differences between the two domains hinder the direct transfer of speculative decoding techniques to image generation. First, the significantly larger sampling space in image generation presents a key challenge for applying speculative decoding. To capture fine-grained visual details, image generation models typically sample 20 to 400 times more candidates from the vocabulary than text models [35, 23, 34], introducing greater variability in token prediction. This makes it much harder for the draft model to align with the target model, limiting the effectiveness of speculative decoding when applied directly.

From another perspective, unlike sequential text, images are naturally two-dimensional, exhibiting complex spatial dependencies that are largely absent in text generation. Our attention-sinking experiments [47] in Section 4.1 reveal that the generation of a pixel-level token is influenced not only by its horizontal neighbors within the same row, but also by vertical dependencies across rows. These findings suggest that both horizontal and vertical spatial contexts provide informative signals, which can be effectively leveraged to guide and accelerate AR image generation. However, existing speculative decoding algorithm designed for text tends to overlook the potential vertical information, and are not well suited to capture two-dimensional spatial context.

To address the aforementioned challenges, we introduce Hawk, a new framework based on Spatial Speculative Decoding that leverages the two-dimensional spatial structure of images to accelerate autoregressive (AR) image generation. By explicitly incorporating spatial dependencies—both horizontal and vertical—Hawk significantly improves alignment between the draft and target models. This not only expands the effective sampling space but also enables more informed speculative predictions, leading to faster generation without compromising image fidelity.

Specifically, Hawk introduces a pair of draft heads that independently speculate in both horizontal and vertical directions. Given the inherent asymmetry between these two spatial dimensions, the horizontal and vertical heads produce complementary speculative results, particularly enhancing diversity in edge contours and central image regions. We then integrate these outputs to construct a unified spatial sampling space, which serves as the foundation for candidate token selection. These candidates are subsequently verified by the target model. Empirically, we find that this merged spatial space aligns more closely with the target model's sampling distribution, improving the acceptance rate of speculative tokens. By enabling multiple tokens to be generated and validated in a single pass, Hawk effectively speeds up inference without compromising generation fidelity. Experimental results on multiple text-to-image benchmarks demonstrate a 1.71× speedup over standard AR models. The main contributions of this work are summarized as follows:

- We highlight the importance of accelerating image generation without compromising output quality, and analyze the unique challenges specific to image generation with speculative decoding.
- We propose Hawk, a novel strategy that combines speculative decoding with spatial information derived from the attention mechanism. To the best of our knowledge, this is the first work to leverage spatial information for accelerating autoregressive image generation.
- By integrating optimized Draft Heads and Spatial Speculative Decoding into a large-scale text-to-image model, Hawk achieves a 1.71× speedup without compromising output quality.

## 2 Related Works

**Autoregressive Text-to-Image Generation.** Autoregressive (AR) models were initially developed for time series tasks. However, subsequent research revealed that AR models could also be applied to spatial data, such as image generation [31]. Early works, including PixelRNN [41] and PixelCNN [40], leveraged RNNs and CNNs to perform pixel-by-pixel image inference. However, this approach had its limitations. Later, methods such as vector quantization [11] were introduced to convert images into discrete tokens. Early approaches like VQ-VAE [42] explored this concept, and as the capabilities of autoregressive models became more apparent, research shifted toward developing more advanced models based on the autoregressive framework. Related works include [9, 30, 49].

In the era of large language models, multimodal generation, which combines both text and image generation, has become one of the key capabilities of large models. Chameleon [35] was proposed as a multimodal model capable of generating both images and text. Lumina-mGPT [23] is a fine-tuned model based on Chameleon. However, due to the reliance on AR inference, these methods inherently suffer from slow inference speeds.

**Speculative Decoding.** Speculative Decoding was initially proposed as an optimization strategy [3, 15] and later widely adopted as an acceleration technique in large language model inference [19, 5]. It achieves acceleration while maintaining output quality, making it a promising method for fast inference. Building on the speculative framework, several enhancements have been proposed to address the challenge of difficulty in obtaining drafter models. For instance, Medusa [4] and Eagle [21] introduced mechanisms such as draft heads, enabling the model to perform speculative generation and verification simultaneously within a single inference pass. Additionally, tree-based attention mechanisms [26] have been employed to increase the number of sampling candidates, enhancing the success rate of speculative decoding.

Further advancements include works like Hydra [1] and KOALA [51], which optimize the training of draft heads to enhance speculative decoding performance. Due to the lack of spatial information utilization in image generation, these acceleration algorithms did not achieve the expected performance gains. In our work, we adopt a similar draft head structure to that in Medusa, enhancing its effectiveness by incorporating spatial information.

**Autoregressive Image Generation Acceleration.** The use of speculative decoding to accelerate image generation remains an underdeveloped area, with no standardized methodology across existing works. Different approaches have been proposed in the literature. For example, the ZipAr [14] adopts a non-autoregressive generation strategy, avoiding the traditional raster scan order. In contrast, the LANTERN++ [18] employs a more relaxed speculative decoding scheme, increasing the acceptance probability of the small model's speculative outputs. The SJD [36] integrates speculative sampling with Jacobi decoding to achieve acceleration.

In our Hawk method, we propose a spatial speculative sampling method. While preserving the autoregressive inference paradigm, we adopt the original speculative decoding strategy to maintain sampling performance, thereby achieving acceleration without compromising the model's accuracy.

# 3  Preliminary

**Autoregressive Image Generation.** An Autoregressive (AR) model follows a next-token prediction process, where the prediction of $x_t$ depends only on its preceding tokens $(x_1, x_2, \ldots, x_{t-1})$. The probability of generating a sequence of discrete tokens can be formulated as: $p(x_1, x_2, \ldots, x_T) = \prod_{t=1}^{T} p(x_t \mid x_1, x_2, \ldots, x_{t-1})$.

In the context of image generation, vector quantization is employed to convert an image into a sequence of tokens. Vector quantization consists of an encoder and a quantizer. The encoder maps an image of shape $x \in \mathbb{R}^{h \times w \times 3}$ into a latent representation $z \in \mathbb{R}^{h \times w \times d}$, where $d$ is the dimension of the latent space. Formally, this can be written as $z = E(x)$. In the process of computing the quantizer, we use a pretrained codebook $\mathcal{Z} = \{z_k\}_{k=1}^{K}$, consisting of $K$ elements, each of dimension $d$. The quantizer then searches for the closest codebook vector for each feature element $z_{i,j}$. Specifically, for each feature point, the quantizer computes: $z_q = \arg\min_{z_k} \|z_{i,j} - z_k\|$, where $z_k$ is a vector in the codebook $\mathcal{Z}$. The quantizer's result is then derived by combining the indices of the nearest neighbors in the codebook for each element in the feature map.

Subsequently, 2D $n \times n$ image tokens are reshaped into a 1D sequence of length $n^2$, enabling next-token generation. As a result, AR-based image generation typically follows a raster scan order, which involves sequential inference of the image, row by row and column by column. The most common pattern is from left to right and top to bottom.

**Speculative Sampling.** The core of speculative decoding lies in its use of the speculative sampling strategy to sample and validate the results generated by the draft model. Suppose the target model has a sampling distribution $P_{\texttt{target}}(x|x_{<t})$ and the draft model has a sampling distribution $P_{\texttt{draft}}(x|x_{<t})$ where $t$ represents the current inference step. The process consists of two phases: drafting and verification. In the drafting phase, we use the draft model to generate $\gamma \in \mathbb{Z}^+$ tokens from the draft sampling space $P_{\texttt{draft}}(x_t|x_{<t})$. However, these tokens are not directly reliable. In the verification phase, we use a single forward pass of the target model to verify the speculative tokens $t+1: t+n$ in parallel. For each speculative token, the target model has a probability of $\min\left(1, p_{\texttt{target}}(x)/p_{\texttt{draft}}(x)\right)$ to accept the token. If no speculative tokens are accepted, the target model resamples the token with the probability $p'_{\texttt{target}}(x) = \text{norm}\left(\max(0, p_{\texttt{target}}(x) - p_{\texttt{draft}}(x))\right)$

**Utilization of Draft Heads Structure.** Since speculative sampling relies on a smaller model for speculation, acquiring such a model can be challenging. The smaller model must maintain good consistency with the target model. However, not all models have smaller parameter versions available. In this context, the draft heads structure [4] serves as an alternative to the draft model. Draft heads refer to additional decoding heads incorporated after the final hidden layers to generate speculative outputs. In the Hawk method, we employ spatial draft heads: horizontal draft heads predict the token at position $(T + \text{Horizontal Depth})$, while vertical draft heads predict the token at position $(T + \text{Vertical Depth} \times \text{Image Width})$. Here, $T$ represents the current position in the autoregressive (AR) decoding process, Horizontal Depth refers to the horizontal speculative distance, and Vertical Depth refers to the vertical speculative distance. By leveraging the draft heads structure, we eliminate the need for a separate, smaller speculative model, allowing us to complete both the draft and verification phases in a single forward pass.

**Attention Sinking.** The concept of "attention sinking" was first introduced in the context of large language decoding [47]. In their study, the authors observed that during the inference process of an autoregressive model, the model's attention tends to concentrate on the first token and a few tokens close to the current inference point. This indicates that the first token and its neighboring tokens contain critical information for the model's reasoning. The loss of this information can lead to a significant decline in the model's performance during inference. To better understand the mechanism of attention sinking in image generation, we introduce two key concepts that will be referenced throughout this paper: inference-neighboring points and spatial-neighboring points. Inference-neighboring points are defined as the set of points that are sequentially adjacent to the current point in raster order (e.g. $x_{T-1}, x_{T-2}, x_{T-3}, x_{T-4}$). Spatial-neighboring points, on the other hand, are defined in terms of the geometric structure of the image. These include the points spatially adjacent to the current point, such as those directly above or to the left of it in the image space (e.g. $x_{T-1}, x_{T-\text{Image width}}, x_{T-2}, x_{T-2 \times \text{Image width}}$).

## 4 Spatial Speculative Decoding

In this section, we first analyze the unique characteristics of the attention sinking phenomenon in autoregressive image generation and observe its reliance on spatial information. We then introduce our proposed Hawk method.

### 4.1 Preliminary Experiments

To analyze the differences in inference dependencies between image and text autoregressive generation, we conduct an attention sinking experiment to compare the attention focus of different models during the generation process. This analysis helps us identify potential strategies for inference acceleration. Our experiment compares the average attention logits across different models to highlight their attention patterns during generation. For the Lumina-mGPT model, we generate a $48 \times 48$ image and examine the average attention logits at its central point. For the text model, we use LLaMA 2 [39], instructing it to generate the same number of tokens as the image central point while observing the average attention logits at the most recently generated token. Our findings reveal that, in image generation, attention is not solely concentrated on inference-neighboring points but also extends to Spatial-neighboring points within the image space. Figure 1 illustrates this relationship.

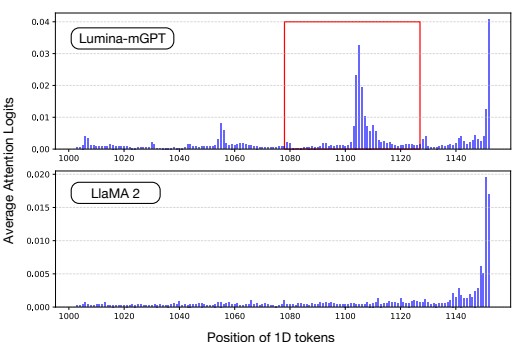

Figure 1: The difference in average attention logits between image generation and text generation, displayed with Lumina-mGPT in a 1D raster order (top), LLaMA 2 (bottom). The average attention logits of the image generation model exhibit a strong dependency on spatially neighboring points. The red box highlights the previous row in the image generation process.

The figure clearly shows that in text generation, attention primarily focuses on inference-neighboring points, with attention scores dropping sharply after approximately five tokens from the current

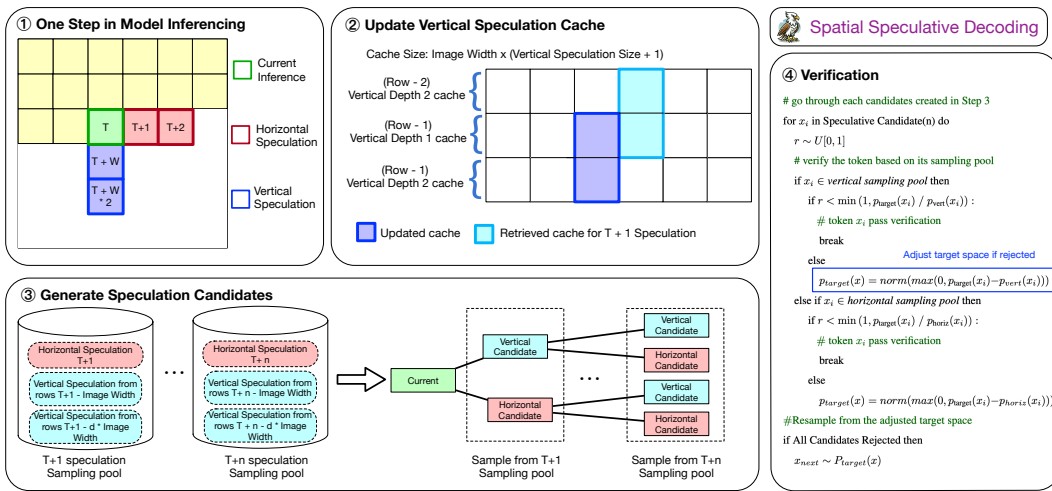

Figure 2: An overview of our Hawk method is presented. During each iteration of the inference process, horizontal and vertical speculations are generated using the draft head. The vertical speculations are stored in the Speculation Cache for future use when processing subsequent lines. Meanwhile, the horizontal speculations are combined with the previous vertical speculations to create the speculation sampling pool. From this pool, tree decoding candidates are generated, followed by a verification step akin to tree speculative decoding.

point. In contrast, for image generation, attention extends beyond the immediate tokens to include spatially adjacent tokens from previous rows, resulting in attention logits with multiple peaks. The red box in the figure highlights the previous row of the image generation. Notably, when inferring the current row, the average attention logits from the previous row surpass those from the current row, suggesting that attention from the preceding row plays a crucial role in image autoregressive inference, even contributing to the majority of the attention. This phenomenon underscores the distinctive dependencies in image generation as compared to text generation.

Speculative sampling employs speculation heads to predict future tokens, but these heads often result in a performance drop compared to the original model's inference. We attribute this performance degradation to insufficient attention information. For text models, key attention information is primarily concentrated around the current inference point. In text-based speculative generation, when attempting to predict tokens farther away, much of the critical attention information is lost, leading to poor performance. In contrast, for image generation models, important information also resides in the spatially adjacent regions around the speculative point. Therefore, even when speculating on distant points (e.g., $T + \text{Image Width}$), the existing inference information from the current row (highlighted by the red box in our attention sinking experiment) helps mitigate the speculation loss. As a result, the draft heads can still maintain relatively good performance despite the absence of some neighboring information. Additional experiments on the attention sinking phenomenon are provided in Appendix B.

## 4.2 Enhancing Speculative Decoding with Spatial Information

**Analysis.** To achieve better speculative inference acceleration, we aim to expand the sampling space of the draft model, increasing its candidate selection options. With a larger sampling space, the likelihood of a smaller model successfully aligning with the sampling distribution of a larger model improves. Simply increasing the number of candidates for speculative decoding in a one-dimensional manner does not inherently enhance sampling diversity, as its effectiveness remains constrained by the limited information available within a single-dimensional space.

Our preliminary experiment demonstrates that directly speculating on tokens below can yield promising results. A potential approach to expand the sampling direction of draft heads is to introduce additional draft heads in speculative sampling, enabling the prediction of the next row or even multiple subsequent rows. By adopting this approach, we effortlessly double the sampling space dimension of the draft model.

Specifically, our strategy consists of three key steps: First, we use dual-direction draft heads to generate an additional sampling space and cache the vertical sample space for use in subsequent inferences. During the speculation process, we combine the cached vertical sampling space from the previous row with the horizontal sampling space at the current speculative point to form a spatial sampling pool, from which we generate our speculative candidates. In this generation process, a tree structure is employed to expand the number of candidates that can be verified simultaneously. Finally, speculative sampling and tree decoding are applied to verify whether the forward pass produces an acceptable speculative result. Figure 2 provides an overview of the method.

**Dual Direction Drafting Heads.** To more effectively explore the target model's sampling space, we believe incorporating spatial information is crucial. We use Dual Direction Drafting Heads to simultaneously speculate in both vertical and horizontal directions, thereby expanding the sampling capacity of our draft heads and improving the success rate of our speculation. Let $T$ represent the current inference position along the horizontal axis (e.g., the current image token in a row). Instead of speculating horizontally on positions $T + 1, T + 2, \ldots, T + n$, where $n$ is the horizontal speculation length, Hawk also makes predictions for positions further down the vertical axis. Specifically, we introduce an additional drafter heads that target positions of the form:

$$T + \text{IW} \times \text{VSD} \tag{1}$$

Where *Vertical Speculation Depth* (VSD) refers to the number of rows below the current position that are explored, and *Image Width* (IW) represents the number of tokens per row in the generated image. In practical usage, we need to follow the raster order for image generation. As a result, the outputs from the vertical draft heads are not utilized immediately; instead, we maintain a cache of size

$$\text{IW} \times \big(\text{VSD} \times (\text{VSD} + 1)\big) \, / \, 2 \tag{2}$$

to store vertically speculative predictions, which will be utilized in the generation of vertical sampling space for subsequent inferences.

**Spatial Sampling Pool.** To optimize our draft sampling space and better align it with the target sampling space, we build a candidate sampling pool using two-dimensional speculating information at each speculation point. We reuse vertical predictions already computed from cache when speculating horizontally to point $T + n$ in the candidates generation step.

When the model computes speculative outputs for $T + n$, it utilizes vertical predictions from up to Vertical Speculation Length subsequent rows at $T + n - IW \times \Delta_v$ position, where $\Delta_v$ represents the subsequent row number. The prediction results at these positions will be combined to form our vertical sampling space.

$$\text{VertSpec}(T + n) = \{\text{Cache}(T + n, \Delta_v, d) \mid d = \Delta_v, \Delta_v \in \{1, 2, \ldots, \text{VSD}\}\} \tag{3}$$

Here, $d$ represents the depth index of the vertical speculation. Specifically, for the vertical speculation results of the previous $k_{th}$ row, we utilize the speculative result with a depth of $k$ at that position. Although these speculative results have different speculation depths and come from different previous rows, all of them point to the same current inference point. These predictions are retrieved from the cache, which we maintained in the previous inference. For constructing the horizontal sampling space, we directly use the speculation from the current point to the $T + n$ point.

$$\text{HorizSpec}(T + n) = P_{\text{horiz}}(x_{T+n} | x \le T) \tag{4}$$

When speculation point $T + n$, we combine the vertical sampling space and the horizontal sampling space to form a two-dimensional spatial sampling pool of next-step candidates. We merge:

$$\underbrace{\text{HorizSpec}(T + n)} \; + \; \underbrace{\text{VertSpec}(T + n)} \tag{5}$$

This merged set forms the new candidate pool at the speculation location $T + n$, incorporating the candidate tree expansion. The candidate tree is generated using the Cartesian product of the sampling results from the sampling pool at each speculation position. By adopting this approach, each newly expanded node in the candidate tree benefits from both the current horizontal speculation and the cached vertical speculation, enabling a more comprehensive exploration of the image space and resulting in richer, more diverse candidate generation.

**Sampling and Verification.** The core principle of speculative decoding lies in preserving the distribution of the original model. Our algorithm focuses on validating this design through a

verification-based decoding process. Specifically, we employ a tree-structured decoding strategy (step-3 in Figure 2) to generate and verify speculative drafts.

For each draft result, candidates at the same depth (e.g., $t+1$, $t+2$, ..., $t+n$) are verified sequentially. If a candidate passes verification, decoding proceeds to the next depth; otherwise, the target distribution is updated iteratively. Specifically, if k candidates are consecutively rejected, we perform k updates:

$$p_{\texttt{target}}^{(i+1)}(x) = \text{norm}\big( \max(0, p_{\texttt{target}}^{(i)}(x) - q_{\texttt{draft}}(x))\big), \quad i = 0, \ldots, k-1, \tag{6}$$

and resample from the last refined distribution $p_{\texttt{target}}^{(k)}$, where $q_{\texttt{draft}}$ may come from either the horizontal or vertical draft head. Despite involving two distinct draft sampling distributions, our approach still preserves the statistical behavior of the original model (see Appendix A for proof). Formally, our final formulation for approximating the original distribution is expressed as:

$$p(x) = \sum_{i=1}^{a+b} \left( \prod_{j=1}^{i-1} (1 - \alpha_j(x)) \cdot q_i(x) \cdot \alpha_i(x) \right) + \left( \prod_{j=1}^{a+b} (1 - \alpha_j(x)) \right) \cdot p_{\text{resid}}(x) \tag{7}$$

where $a$ and $b$ denote the numbers of samples drawn from the two sampling spaces, and $\alpha$ represents the acceptance rate at each verification step.

Here, $p_{\text{resid}}(x)$ denotes the residual probability of failure after all candidates have been evaluated. In conventional speculative sampling, multiple candidates are drawn from a single latent space, yielding identical candidate distributions. As a result, when all candidates fail verification, the residual probability $p_{\text{resid}}(x)$ cannot be further reduced. In contrast, by introducing multiple sampling heads that operate on distinct spaces, our approach diversifies the verification distributions, thereby mitigating the residual probability and improving overall sampling efficiency.

## 5 Experiment

**Model and Benchmark.** We evaluate our proposed Hawk method on the Lumina-mGPT model [23]. For testing, we generate $768 \times 768$ images and use top-k sampling with k = 2000 and a temperature of 1.0. As for how these parameters affect the generated images, we provide a discussion in Appendix D. We also employ classifier-free guidance [8] with a guidance scale of 3.0. For the benchmark, we use the test sets from COCO 2017 [22] and Flickr30K [29]. From these datasets, we sample 500 examples from each for evaluation.

**Metrics.** We evaluate inference acceleration using two metrics: Inference Acceleration Rate (*original inference time / accelerated inference time*), where higher is better, and Accept Length, the average number of successful speculative predictions per step. For non-speculative baselines, the accept length is 1. To evaluate image quality, we employed two widely used metrics. The FID [17] score measures the similarity between the generated images and real images, where lower scores indicate better quality. Additionally, the CLIP [16] score assesses the semantic alignment between the generated images and their corresponding text prompts, with higher scores reflecting better alignment. All experiments are conducted on RTX 3090 GPUs.

**Training.** To train our Spatial Draft Head, we largely follow the same procedure used for Medusa and Lumina-mGPT. We only train the draft head, while keeping the rest of the model frozen. We use the AdamW [24] optimizer with a weight decay of 0.1 and $\beta = (0.9, 0.95)$. The base learning rate is set to $2 \times 10^{-5}$. We set the draft head balance weight $\lambda_k$ to 1. Structural details of the draft heads and memory considerations are provided in Appendix C. We use $6,000$ images sampled from the LAION aesthetic training set [32] to fine-tune our drafter head. This training procedure typically takes 8–12 hours to converge on a **single** RTX 3090 GPU.

### 5.1 Quantitative and Qualitative Evaluation

We evaluate five different methods by analyzing their performance across various test datasets: vanilla autoregressive decoding, Medusa (horizontal draft heads), Hawk with vertical draft heads, Medusa with LANTERN++, and Hawk with spatial draft heads. In this context, spatial draft heads refer to the combination of both vertical and horizontal draft heads.

Table 1: Experiment result for different speculation methods on COCO2017 and Flickr30K validation dataset. We sampled 500 images from the validation set and evaluated the hardware environment of a single RTX 3090 GPU. Spatial Draft heads: a combination of vertical draft heads and horizontal draft heads. For LANTERN++, we use the hyperparameter setting of $(k = 10, \ \lambda = 2)$.

| Dataset | Method | Speed up | Accepted Length | FID | CLIP Score |
|---------|--------|----------|-----------------|-----|------------|
| COCO2017 | Vanilla AR | 1.00× | - | 90.68 | 33.43 |
| | Medusa [4] | 1.58× | 1.728 | 92.51 | 33.34 |
| | Hawk (Vertical Draft Heads) | 1.58× | 1.726 | 91.46 | 33.33 |
| | LANTERN++ [18] | 1.69× | 1.863 | 93.22 | 33.08 |
| | **Hawk (Spatial Draft Heads)** | **1.71×** | **1.890** | **90.71** | **33.39** |
| Flickr30K | Vanilla AR | 1.00× | - | 104.80 | 34.49 |
| | Medusa [4] | 1.55× | 1.704 | 103.51 | 34.48 |
| | Hawk (Vertical Draft Heads) | 1.59× | 1.738 | 104.21 | 34.45 |
| | LANTERN++ [18] | 1.67× | 1.842 | 106.10 | 34.25 |
| | **Hawk (Spatial Draft Heads)** | **1.69×** | **1.871** | **104.50** | **34.45** |

Table 1 presents the quantitative results comparing different approaches in the image speculative decoding task. Our vertical and horizontal draft heads demonstrate varying effectiveness depending on the dataset. This discrepancy is likely attributed to the distinct image distribution characteristics of each dataset. Due to varying spatial dependencies within the images across datasets, performance differences arise between the different draft heads. Our Hawk method achieves a $1.71\times$ speed-up compared to the baseline method, aligning with our expectations. Since our sampling space is a combination of vertical and horizontal sampling spaces, its overall performance is constrained by the summation of the performance gains from both heads.

From the perspective of FID (lower is better) and CLIP score (higher is better), our approach achieves substantial acceleration while effectively preserving the image quality of the original model. We compare our method with LANTERN++, an existing multi-draft image speculative decoding acceleration strategy, and integrate their algorithm into our Medusa-based framework for a fair comparison. Our approach achieves a consistently higher accepted length during speculative verification, indicating more efficient and stable speculative decoding. Although the overall acceleration gain is moderate due to the additional computation from dual-directional draft heads, this improvement is obtained without relaxing verification thresholds, maintaining strict adherence to the verification criteria.

In contrast, LANTERN++ adopts a relaxed speculative decoding strategy from $\min \left(1, \frac{q(\hat{x}|s)}{p(\hat{x}|s)}\right)$ to $\min \left(1, \frac{\sum_{x \in A_k, \, \delta(\hat{x})} q(x|s)}{p(\hat{x}|s)}\right)$ that substantially lowers the verification threshold, increasing the acceptance rate of inaccurate predictions from the small model. This often results in degraded generation quality, as reflected in the fact that LANTERN++ exhibits higher FID and lower CLIP scores compared to Hawk. By maintaining a more conservative and accurate verification process, Hawk achieves superior overall performance in both efficiency and output fidelity.

For the qualitative evaluation, we visually compared images generated by different methods to identify any noticeable differences in quality. This analysis aimed to assess whether the accelerated methods introduced any degradation compared to the baseline or other approaches. Given that our speculative sampling strategy is mathematically grounded, it theoretically preserves the same sampling space as the target model. As shown in Figure 3, we conducted extensive observational experiments on our test set and found no significant degradation in image quality. These results suggest that our Hawk method produces images that maintain the same quality as the original model.

## 5.2 Effectiveness of Vertical Draft Head

We aim to demonstrate that, compared to traditional horizontal speculative sampling, vertical heads introduce new information that improves the spatial sampling space. We tested performance differences between vertical and horizontal draft heads during training and inferencing.

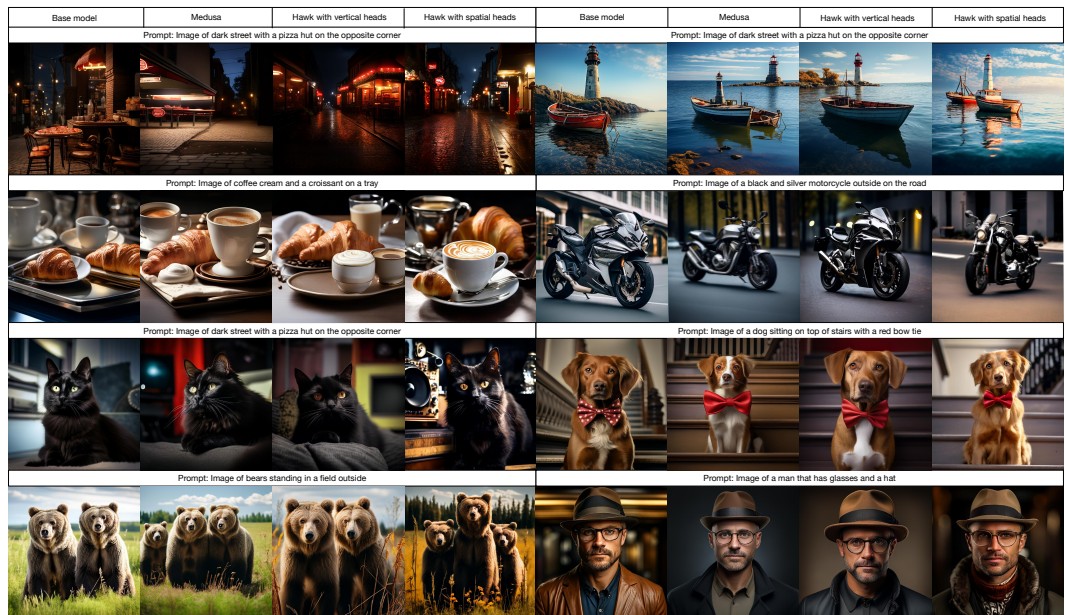

Figure 3: Quality evaluation of Hawk method. For each image type, the images, from left to right, are generated by the base model, Medusa, Hawk with vertical heads, and Hawk with spatial heads. The Hawk method maintains the performance of the baseline model while enhancing inference speed.

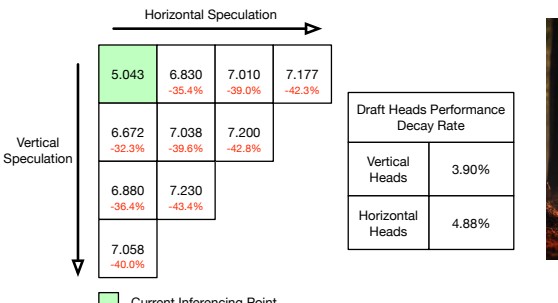

Figure 4: The training loss of draft heads at different locations is related to the current decoding point. Vertical heads experience relatively less performance decay as the speculation depth increases.

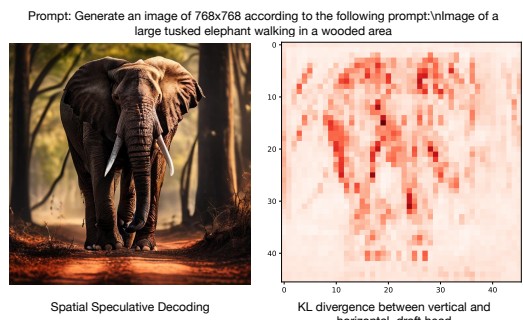

Figure 5: The KL divergence between the vertical and horizontal draft heads during speculative decoding for the given prompt. The difference between the vertical and horizontal draft heads is more pronounced when generating complex areas of the image.

During training, we observed a discrepancy in the speculative capabilities of vertical and horizontal heads. We compared the training losses of draft heads at different positions. Intuitively, since the vertical speculation head is positioned at a distance equal to the image width from the current decoding point—much larger than that of the horizontal head—we would expect the vertical draft head to have a higher training loss. However, our experiments yielded the opposite result. In Figure 4, we show comparisons of training losses of draft heads at various positions.

The green sections represent the fine-tuning loss of the model on the training dataset, while the other sections show the training loss of the speculative draft heads. For speculative points at the same Euclidean distance from the current decoding position, the vertical draft head demonstrates lower training loss. The performance decay rate shown in the figure illustrates this phenomenon. With increasing speculative depth, the vertical direction shows a smaller performance decay rate (3.90%) compared to the horizontal direction (4.88%). Our experiments reveal that, ideally, vertical heads should produce better speculative results than horizontal heads, even though they involve a significantly larger speculative distance.

In Figure 5, we compare the Kullback–Leibler (KL) divergence between horizontal and vertical draft heads during inferencing. The plot reveals that the divergence gap becomes more pronounced when generating fine-grained details, such as an elephant's eyes, nose, teeth, and overall contour. These regions, due to their inherent ambiguity, permit multiple potential outcomes, resulting in a broader sampling space for the target model and greater KL differences between vertical and horizontal heads. Conventional draft-based speculative sampling struggles to align with the target model in such cases. In contrast, our Hawk method leverages integrated sampling spaces, with additional contributions from the vertical head, which broadens the range of speculative choices and significantly increases the acceptance rate of our sampling strategy.

To further quantify the benefits of this increased diversity among draft heads, we analyze its theoretical impact on sampling efficiency. As shown in Figure 6, we illustrate the theoretical speedup effects introduced by the diversity among different draft heads. We evaluate the theoretical speculative rejection probability $p_{\text{resid}}(x)$ (see Eq. 7), where a lower $p_{\text{resid}}(x)$ indicates greater potential for acceleration. The term $p_{\text{resid}}(x)$ represents the residual probability of resampling after rejection.

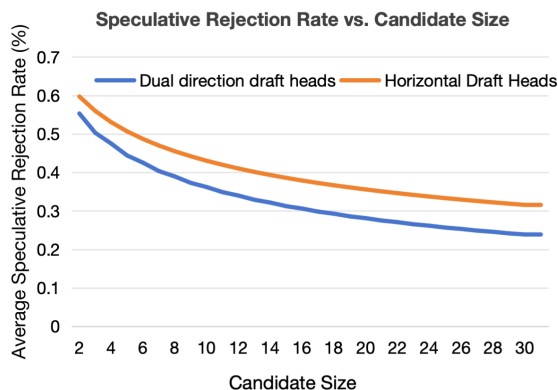

Figure 6: Comparison of dual-direction and horizontal draft heads across varying numbers of candidates. The dual-direction design yields a lower speculative decoding rejection probability, indicating more efficient utilization of draft tokens.

As defined in Eq. 6, the residual distribution $p_{\text{target}}^{(i+1)}(x)$ is recursively updated by subtracting the draft distribution $q_{\text{draft}}(x)$ from the target distribution. When the same draft distribution $q_{\text{draft}}(x)$ is repeatedly used across iterations, the residual mass converges to a fixed point where further subtraction yields no meaningful reduction—hence $p_{\text{resid}}(x)$ cannot further converge. However, by introducing multiple sampling spaces with distinct draft distributions, the residuals at each step are computed with respect to different $q_{\text{draft}}^{(j)}(x)$, enabling additional reduction of the overall $p_{\text{resid}}(x)$ and thus yielding greater theoretical speedup.

## 6 Conclusion

In this work, we introduced Hawk, an acceleration framework for autoregressive image generation that leverages spatially aware speculative decoding. By integrating spatial information into the sampling process, Hawk effectively enlarges the candidate space and achieves substantially higher acceptance rates, leading to faster inference without compromising image quality. Extensive experiments demonstrate that Hawk maintains visual fidelity comparable to the baseline while delivering notable speedups. Future work includes integrating Hawk with more advanced speculative backbones (e.g., Eagle [21] or Hydra [1]) to further enhance efficiency and performance.

## 7 Acknowledgements

This work was supported by the National Key R&D Program of China (No. 2024YFE0202800), the National Natural Science Foundation of China (NSFC) under Grant No. 62476123, and NSFC under Grant No. 62376118.

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

## A Theoretical Justification

Assume we are given two speculative sampling spaces: one from the vertical draft head and the other from the horizontal draft head. Let $a$ denote the number of speculative tokens sampled from the vertical head, and $b$ denote the number of speculative tokens sampled from the horizontal head. In the simplest case, we set $a = 1$ and $b = 1$, meaning that we draw one candidate token from each head for validation. We adopt a sequential validation strategy in which the token proposed by the vertical head is evaluated first. If it is rejected, we then proceed to validate the token from the horizontal head. Under this setup, we define the following notation: $p(x)$ be the true distribution defined by the base AR model. $q_{\text{vert}}(x)$ and $q_{\text{horiz}}(x)$ be the proposal distributions from the vertical and horizontal draft heads, respectively.

$p'_{\text{vert}}(x)$ is the adjusted base model distribution after vertical head rejection. Calculated by

$$\text{norm}(\max(0,\, p(x) - q_{\text{vert}}(x))) = \frac{p(x) - \min(q_{\text{vert}}(x),\, p(x))}{\sum_{x'} [p(x') - \min(q_{\text{vert}}(x'),\, p(x'))]} \tag{8}$$

$p'_{\text{vert-horiz}}(x)$ is the residual distribution after both vertical and horizontal rejections. Calculated by

$$\text{norm}(\max(0,\, p'_{\text{vert}}(x) - q_{\text{vert}}(x))) = \frac{p'_{\text{vert}}(x) - \min(q_{\text{vert}}(x),\, p'_{\text{vert}}(x))}{\sum_{x'} [p'_{\text{vert}}(x') - \min(q_{\text{vert}}(x'),\, p'_{\text{vert}}(x'))]} \tag{9}$$

$\alpha_{\text{vert}}$ is the acceptance probability of the vertical head. Calculated by

$$1 - \sum_{x'} (p(x') - \min(q_{\text{vert}}(x'), p(x'))) \tag{10}$$

$\alpha_{\text{vert-horiz}}$ is the acceptance probability of the horizontal head conditioned on vertical rejection. Calculated by

$$1 - \sum_{x'} (p_{\text{vert}}(x') - \min(q_{\text{vert}}(x'), p'_{\text{vert}}(x'))) \tag{11}$$

Our ultimate goal is to approximate the original model's distribution p(x) by combining samples from two draft heads operating along different spatial dimensions. For a given point x', our proof proceeds as follows:

$$
\begin{aligned}
p(x') &= \Pr(\text{vertical accept},\ x = x') + \Pr(\text{vertical reject, horizontal accept},\ x = x') \\
&\quad + \Pr(\text{vertical reject, horizontal reject},\ x = x') \\
&= q_{\text{vertical}}(x') \cdot \min\left(1, \frac{p(x')}{q_{\text{vertical}}(x')}\right) \\
&\quad + (1 - \alpha_{\text{vertical}}) \cdot q_{\text{horizontal}}(x') \cdot \min\left(1, \frac{p'_{\text{vertical}}(x')}{q_{\text{horizontal}}(x')}\right) \\
&\quad + (1 - \alpha_{\text{vertical}})(1 - \alpha_{\text{vertical-horizontal}}) \cdot p'_{\text{vertical-horizontal}}(x') \\
&= q_{\text{vertical}}(x') \cdot \min\left(1, \frac{p(x')}{q_{\text{vertical}}(x')}\right) \\
&\quad + (1 - \alpha_{\text{vertical}}) \cdot q_{\text{horizontal}}(x') \cdot \min\left(1, \frac{p'_{\text{vertical}}(x')}{q_{\text{horizontal}}(x')}\right) \\
&\quad + (1 - \alpha_{\text{vertical}}) \cdot \left(p'_{\text{vertical}}(x') - \min(q_{\text{horizontal}}(x'),\, p'_{\text{vertical}}(x'))\right) \\
&= q_{\text{vertical}}(x') \cdot \min\left(1, \frac{p(x')}{q_{\text{vertical}}(x')}\right) + (1 - \alpha_{\text{vertical}}) \cdot p'_{\text{vertical}}(x') \\
&= p(x') \tag{12}
\end{aligned}
$$

Let $\pi = [s_1, s_2, \ldots, s_{a+b}]$ denote an ordered sequence of speculative tokens, where each $s_i$ is generated from either a vertical or horizontal draft head. The total number of speculative tokens is $a + b$, with $a$ sampled from horizontal heads and $b$ from vertical heads. Each sample result $s_i$ corresponds to a proposal posbility $q_i(x)$. At each step i, the acceptance probability is defined

as $\alpha_i = 1 - \sum_{x'} \left(p^{(i)}(x') - \min(q_i(x'), p^{(i)}(x'))\right)$ where $p^{(i)}(x)$ represents the residual target density after the first $i - 1$ proposals have been rejected. It is recursively defined as: $p^{(i)}(x) = \text{norm}\left(\max\left(0, , p^{(i-1)}(x) - q_{i-1}(x)\right)\right)$, The sampling probability at a specific point $x = x'$ under the generalized formulation is given by:

$$p(x) = \sum_{i=1}^{a+b} \left(\prod_{j=1}^{i-1} (1 - \alpha_j(x)) \cdot q_i(x) \cdot \alpha_i(x)\right) + \left(\prod_{j=1}^{a+b} (1 - \alpha_j(x))\right) \cdot p_{\text{resid}}(x) \qquad (13)$$

This section provides the theoretical justification needed to address the reviewer's concern about potential distributional shift. It shows that our dual-direction speculative decoding preserves the original output distribution.

## B   More Attention Sinking Experiment

We conducted further analysis of attention sinking in image generation. In Figure 7, we reshaped the 1D image attention logits into a 2D spatial representation for visualization. The results reveal that during the generation of an image, significant attention is concentrated on the first token of the image and the tokens spatially adjacent to the current inference point. Additionally, tokens at the end of each row tend to have slightly higher weights, while the rest of the attention is evenly distributed across the entire image. Although image generation still follows a raster order, it inherently encodes spatial information.

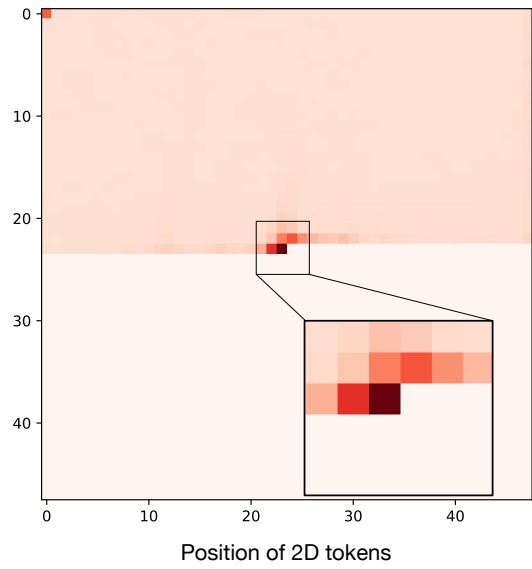

Position of 2D tokens

Figure 7: Lumina-mGPT average attention logits converted into a 2D spatial representation.

## C   Draft Heads Structure and Memory Overhead

For a single Medusa head, we employ a linear layer with both input and output dimensions equal to the model's hidden size, followed by a SiLU activation function to introduce non-linearity.

The original Medusa strategy introduces only minimal memory overhead (less than 0.5%). Our proposed SSD strategy results in a relatively small increase in the total number of parameters, accounting for less than 1% of the base model (Lumina-mGPT). During inference, the actual memory overhead ratio is even lower due to the use of KV caching, which helps reduce the memory cost of additional components. In the case of FP16, our draft heads occupy approximately 134 MB of memory.

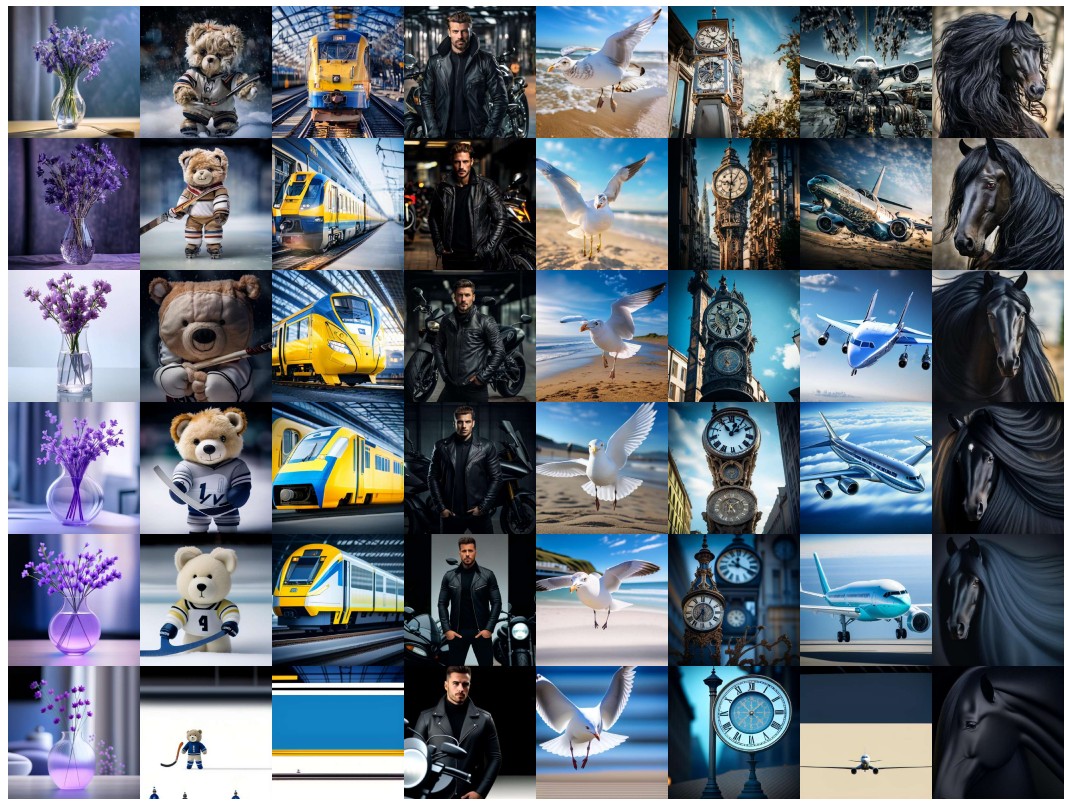

Figure 8: The performance of the base model under different top-k parameters is as follows: from top to bottom, the values of top-k are 2000, 1000, 200, 50, 10, and 1.

## D  Different Generation Parameters Affect Image Quality

In AR text generation, the top-k parameter is typically set between 5 and 50, representing the maximum number of elements considered during sampling at each step. However, in AR image generation, the choice of top-k has a significantly different impact. Larger top-k values generally result in more detailed images and higher image quality. However, as top-k increases, speculative sampling becomes more challenging: fewer speculative predictions pass the model's validation, leading to slower speculative decoding speeds. Figure 8 compares how different top-k values affect the image quality of the baseline model.

Temperature plays a crucial role in autoregressive image generation, as the quality of the generated images varies significantly with changes in temperature. In Figure 9, we tested the performance of the base model under different temperature settings. Due to the strong impact of temperature on image quality, certain algorithms, such as jacobi decoding [27] (which is only applicable in greedy decoding), are not suitable for accelerating autoregressive image generation.

## E  Limitation and Future works

In our current implementation, we employ Medusa as the baseline for speculative decoding. However, according to prior evaluations [46], Medusa is not among the most efficient or state-of-the-art algorithms for text speculative decoding. Consequently, our reported acceleration results may underestimate the potential performance achievable with more advanced speculative decoding methods. In future work, we plan to integrate stronger baseline algorithms to further improve inference efficiency and validate the generality of our proposed approach.

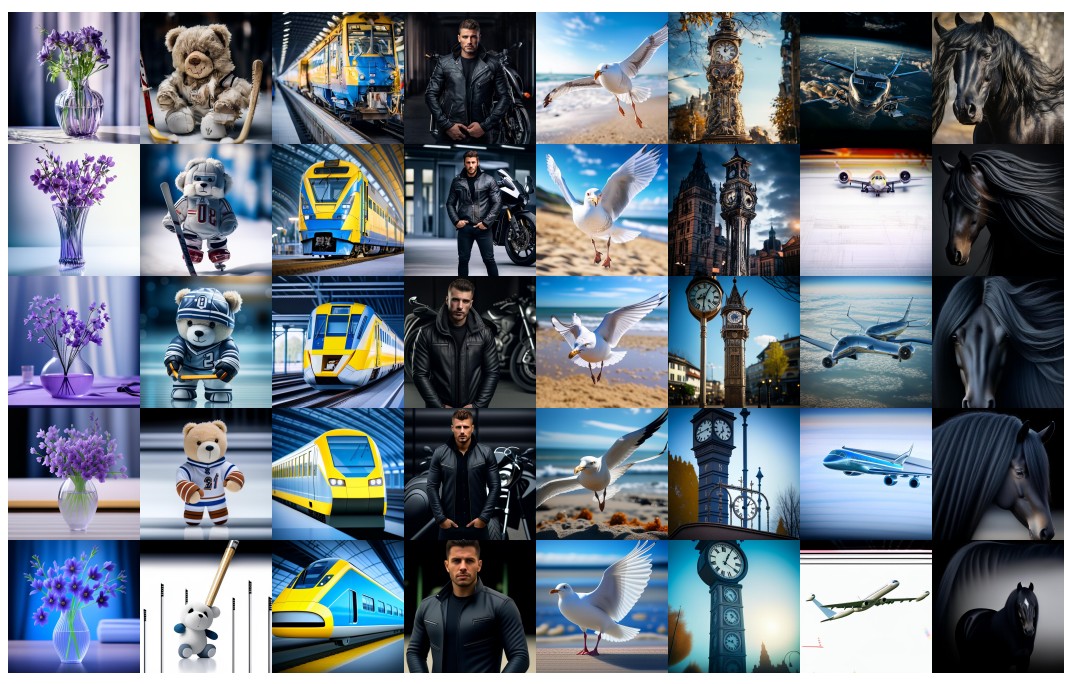

Figure 9: The performance of the base model under different temperature parameters is as follows: from top to bottom, the values of temperature are 1, 0.8, 0.6, 0.4, 0.2.

