# OpenReview forum: "Hawk: Leveraging Spatial Context for Faster Autoregressive Text-to-Image Generation"
_NeurIPS.cc/2025/Conference — NeurIPS 2025 poster_

### Official Review · Reviewer_httE · 2025-06-30

**Clarity:** 3
**Significance:** 3
**Originality:** 3
**Rating:** 4
**Confidence:** 4

**Summary:**

This paper introduces Hawk, a novel approach designed to accelerate autoregressive text-to-image generation by leveraging the two-dimensional spatial structure inherent in images. The core idea is to enhance speculative decoding by incorporating both horizontal and vertical spatial information, which helps improve the alignment between draft and target models, leading to faster inference without compromising image quality.

**Questions:**

1. The fundamental concept presented in this paper is highly dependent on raster order generation. This raises a question about whether the observed attention sink exists in other non-raster autoregressive generation order.

**Ethical Concerns:**

["NO or VERY MINOR ethics concerns only"]

**Final Justification:**

The author addressed my concerns in their rebuttal and I maintain my positive rating.

**Limitations:**

Yes

**Quality:**

4

**Strengths And Weaknesses:**

Strengths:

1. The paper proposes a novel method, Hawk, which is the first to leverage two-dimensional spatial information for accelerating autoregressive image generation using speculative decoding.
2. Hawk demonstrates a notable 1.71x speedup over standard AR models on multiple text-to-image benchmarks (COCO2017 and Flickr30K), while effectively preserving both image fidelity (lower FID scores) and diversity (higher CLIP scores). The quantitative results presented in Table 1 provide clear evidence of its performance.

Weakness:

1. It is advisable to display the acceptance rate at each location, similar to Figure 4. This will provide a more straightforward illustration of the effectiveness of the proposed method.
2. Formatting questions: the equations in this paper lack tags, which makes it difficult to locate the corresponding equations in the context.

---

> ### Author Rebuttal · Authors · 2025-07-31
>
> We sincerely thank the reviewer for recognizing the timeliness and importance of the problem we aim to address, as well as for acknowledging the strong empirical performance of our proposed method. We deeply appreciate your constructive feedback, which has helped us identify several areas for improvement.
>
> In response to your suggestion on visualizing spatial acceptance behavior, we fully agree that examining acceptance length across different spatial locations can provide valuable insights into the dynamics of our acceleration process. Accordingly, we plan to include an acceptance heatmap in the “Additional Experimental Results” section of the final version, which will visually demonstrate how acceptance varies across image regions and reinforce the effectiveness of our method. Regarding the formatting issue of missing equation numbers, we apologize for the oversight. We will ensure that all mathematical expressions are properly tagged in the camera-ready version to improve clarity and ease of reference.
>
> In response to your question about whether the attention sink phenomenon exists in non-raster autoregressive generation orders (which we interpret as generation paradigms that produce complete images progressively rather than token-by-token in raster order, such as VAR or ViDT), we note that similar attention behaviors have indeed been reported. For example, the paper Analysis of Attention in Video Diffusion Transformers observes attention sinking in ViDT models. However, these patterns exhibit some differences from those in traditional autoregressive models. Specifically, attention sinks in ViDTs are typically (1) **spatially random**—tokens associated with sink behavior are scattered rather than clustered in semantically significant regions, and (2) **temporally biased**—they appear more frequently in early latent frames and gradually diminish in later ones. In other words, they show temporal regularity but lack consistent spatial structure. To the best of our knowledge, while attention sinking has been observed in non-raster generation settings, no existing acceleration method has systematically leveraged this behavior for inference speedup. We agree that this is a compelling and underexplored direction, and we thank the reviewer for bringing it to our attention.
>
> Once again, we sincerely thank the reviewer for recognizing the significance of our problem and for providing constructive and actionable feedback. We warmly welcome any additional questions or suggestions you might have, and would be happy to provide further clarification.

---

> > ### Comment · Reviewer_httE · 2025-08-04
> >
> > Thanks for the author's rebuttal. The authors address to my question about whether the observed attention sink exists in other non-raster autoregressive generation order. Therefore, I maintain my original opinion.

---

> > > ### Author Response · Authors · 2025-08-05
> > >
> > > Thank you very much for your thoughtful review and for taking the time to engage with our rebuttal. We’re pleased that our response addressed your question regarding attention sink behaviors in non-raster generation orders, and we sincerely appreciate your continued support. We are also grateful for your recognition of our contributions, and we will ensure that your suggestions are reflected in the final version of the paper.

---

### Official Review · Reviewer_LJzH · 2025-07-01

**Clarity:** 2
**Significance:** 2
**Originality:** 2
**Rating:** 4
**Confidence:** 4

**Summary:**

This paper addresses the slow inference issue in autoregressive (AR) text-to-image generation models, noting that while speculative decoding accelerates text generation, its application to images is underexplored due to larger sampling spaces and unexploited 2D spatial dependencies. The authors propose Hawk, a framework that leverages horizontal and vertical spatial contexts via dual draft heads, constructing a unified sampling pool by merging horizontal speculations with cached vertical predictions. Empirically, Hawk achieves a 1.71× speedup over standard AR models on benchmarks like COCO2017 and Flickr30K, preserving image fidelity and diversity.

**Questions:**

See Weaknesses

**Ethical Concerns:**

["NO or VERY MINOR ethics concerns only"]

**Final Justification:**

I thank the authors for the rebuttal. Most of my concerns have been addressed. Hence, I decide to increase my score to 4.

**Limitations:**

See Weaknesses

**Quality:**

2

**Strengths And Weaknesses:**

Strengths:
1. The problem is timely and important.
2. The empirical performance seems good.

Weaknesses:
1. This paper does not consider state-of-the-art Autoregressive (AR) image generation models, e.g., VAR. Moreover, the proposed technique cannot be applied to other types of generation models, such as diffusion models, flow-matching, and consistency models.
2. The proposed technique does not introduce many innovations on top of the vanilla version of speculative decoding (the current 1D --> 2D may not be novel enough).
3. Given 2, the experiments are not extensive enough. The authors should consider state-of-the-art generative fundation models learned on significantly larger datasets (e.g., at least at the level of Stable Diffusion).

---

> ### Author Rebuttal · Authors · 2025-07-31
>
> We sincerely thank the reviewer for the thoughtful and constructive feedback. We appreciate your recognition of both the problem’s significance and our empirical results. Below, we would like to clarify a few key aspects of our setting, comparison scope, and technical contributions that may not have been fully explained in the original submission.
>
> ### 1. Clarification on Comparison Scope:
>
> Regarding the comment:
> “This paper does not consider state-of-the-art Autoregressive (AR) image generation models,”
>
> We believe the apparent discrepancy may stem from **differing assumptions about the target setting**. While VAR focuses on **autoregressive image generation in isolation**, our work is situated in a different and increasingly popular setting: **unified multimodal foundation models** that support next-token generation across both text and image modalities.
>
> This line of research has recently attracted substantial interest in the multimodal community, as it enables interleaved text-image generation through a unified autoregressive backbone. Representative examples include Chameleon[6], Lumina-mGPT[10,11], BLIP3-o[12], Emu3[13], Bagel[14], and Uni-World[15], all of which follow similar generation pipelines.
>
> In this setting, the typical generation flow involves:
>
> 1. An LLM performs autoregressive inference over the text portion.
> 2. **The model continues autoregressively to generate image tokens (often in a raster order).**
> 3. A decoder, such as a VAE or diffusion model, reconstructs the final image from these tokens.
>
> Our work specifically targets the **second step**, which is often the **computational bottleneck**, and aims to **accelerate the token-level generation process within these foundation models**. Importantly, models like VAR[16], Diffusion, or Consistency Models are used primarily for decoding images in the **final step** and are not responsible for autoregressive token generation in the latent space. As such, they operate in a **different part of the pipeline** and are not the focus of the acceleration we propose.
>
> This evaluation practice is consistent with recent works that address the same generation setting—such as LANTERN++[1] (ICLR 2024), SJD[2] (ICLR 2024), and  ZiPar[3] (cvpr)—which also focus on token-level generation and do not include VAR as a baseline due to the fundamental differences in modeling scope.
>
> ### 2. Clarification on Application Scope:
>
>
> Regarding the comment: “The proposed technique cannot be applied to other types of generation models, such as diffusion models, flow-matching, and consistency models.”
>
> We sincerely thank the reviewer for raising this important point. Indeed, we fully agree that our proposed method is tailored to the autoregressive (AR) generation paradigm, and it may not be directly applicable to other generation frameworks such as diffusion models, flow-matching, or consistency models. This is **not a limitation unique to our work**, but rather reflects **a broader landscape** where **acceleration techniques are often specialized to the structural properties of the target model class**.
>
> As also noted in recent surveys [4,5], acceleration methods for AR transformers and diffusion models tend to **evolve independently**, given their **fundamentally different generation processes**—sequential token prediction vs. iterative denoising. For instance, techniques such as Speculative Diffusion Decoding [8] and Cache Me If You Can [9] are powerful in the diffusion setting but are tightly coupled with the nature of score-based or noise-conditioned sampling. These cannot be directly transferred to token-level AR models.
>
> Similarly, our method—like prior AR acceleration work including LANTERN++ [1], SJD [2], and Zipar [3]—is specifically designed to leverage the autoregressive sampling structure. While we believe bridging the gap across paradigms is a compelling future direction, it would likely require fundamentally new algorithmic tools beyond the scope of the current work.
>
> We hope this clarification helps contextualize our design decisions, and we truly appreciate the reviewer’s thoughtful feedback on this point.
>
> ### 3. Clarifying the Novelty Beyond 1D-to-2D Extension
>
> Regarding the comment: “The proposed technique does not introduce many innovations on top of the vanilla version of speculative decoding (the current 1D --> 2D may not be novel enough)”
>
> Thank you very much for your thoughtful feedback. We understand your concern that the transition from 1D to 2D speculative decoding may seem incremental at first glance. However, we respectfully argue that adapting speculative decoding to the two-dimensional structure of image generation is **non-trivial both algorithmically and conceptually**, and introduces several new challenges that we address in our work.
>
> Unlike prior speculative decoding approaches that operate on linear (1D) sequences, our method is designed specifically for 2D spatial data, where information flows both horizontally and vertically. To effectively handle this structure while preserving the autoregressive (AR) generation order, we propose several key innovations:
>
> - We design **vertical draft heads** to enable speculative decoding along the vertical axis, which is entirely absent in prior methods.
>
> - We introduce a **spatial cache mechanism** that stores vertical speculative results across multiple future rows, allowing reuse of these intermediate computations **without violating the AR generation order**.
>
> - We further propose a **fused speculative sampling algorithm** that jointly leverages both cached vertical information and horizontal predictions, **balancing efficiency with consistency** in the 2D decoding trajectory.
>
> We believe these contributions go beyond a simple dimensional extension and represent a meaningful step toward adapting speculative decoding to structured generation domains such as vision. Moreover, several reviewers have recognized the novelty of this contribution:
>
> - **Reviewer g8rp** noted that our method “employs dual-direction draft heads and spatial caches that explicitly align with the two-dimensional image structure, offering **an interesting way to leverage spatial dependencies.**”
>
> - **Reviewer httE** described it as “**the first** to leverage two-dimensional spatial information for accelerating autoregressive image generation using speculative decoding.”
>
> - **Reviewer UpDY** commented that our method “presents **a new speculative decoding method** considering the two-dimensional spatial context of images.”
>
> We are grateful that these reviewers found our contributions promising, and we hope that this clarification better highlights the novelty and technical depth of our proposed method. We also believe this work may open new directions for applying speculative decoding in other structured domains beyond language.
>
> ### 4. Clarifying the Choice of Foundation Models
>
> Regarding the  comment: “ The authors should consider state-of-the-art generative fundation models learned on significantly larger datasets (e.g., at least at the level of Stable Diffusion)”
>
> To clarify, the foundation model we use, Lumina-mGPT [10,11], is actually a derivative of Chameleon [6], a large-scale vision-language model developed by Meta. Since Meta has not released the VAE component (which handles image tokenization), the Lumina-mGPT project was created to fill in that missing piece and make the model accessible for academic use. That said, most of the model’s parameters are directly inherited from Chameleon, and only the VAE was newly trained.
>
> In terms of data scale, we’d like to point out that Lumina-mGPT was trained on about **1.9 trillion image-text tokens**. For comparison, Stable Diffusion 3.0[7], the latest version based on the Stable Diffusion Transformer architecture, was trained on roughly **1 billion image-text pairs**. The exact number of tokens may vary depending on the tokenizer used. While the data formats differ, we believe it is difficult to say that SD3.0 was trained on significantly more data, especially when considering the total token volume.
>
> Finally, we’d like to mention that **Lumina-mGPT is increasingly being used in the research community** as a practical foundation model. Two recent papers with settings similar to ours, SJD (ICLR) and LANTERN++ (ICLR), both adopt Lumina-mGPT as their main base model.
>
> We respectfully hope that these clarifications provide additional context for our design choices and contributions. We remain open to further discussion and sincerely appreciate the opportunity to engage with your valuable feedback.
>
> Reference
>
> [1]LANTERN++: ENHANCING RELAXED SPECULATIVE DECODING WITH STATIC TREE DRAFTING FOR VISUAL AUTO-REGRESSIVE MODELS
>
> [2]Accelerating Auto-regressive Text-to-Image Generation with Training-free Speculative Jacobi Decoding
>
> [3]ZipAR: Parallel Auto-regressive Image Generation through Spatial Locality
>
> [4]A Comprehensive Survey of Accelerated Generation Techniques in Large Language Models
>
> [5]Efficient Diffusion Models: A Comprehensive Survey from Principles to Practices
>
> [6]Chameleon: Mixed-Modal Early-Fusion Foundation Models
>
> [7]Scaling Rectified Flow Transformers for High-Resolution Image Synthesis (stable diffusion 3.0)
>
> [8] Speculative Diffusion Decoding: Accelerating Language Generation through Diffusion
>
> [9] Cache Me if You Can: Accelerating Diffusion Models through Block Caching
>
> [10]Lumina-mGPT: Illuminate Flexible Photorealistic Text-to-Image Generation with Multimodal Generative Pretraining
>
> [11]Lumina-mGPT 2.0:Stand-Alone AutoRegressive Image Modeling
>
> [12]BLIP3-o: A Family of Fully Open Unified Multimodal Models-Architecture, Training and Dataset
>
> [13]Emu3: Next-Token Prediction is All You Need
>
> [14]Emerging Properties in Unified Multimodal Pretraining (Bagel)
>
> [15]UniWorld-V1: High-Resolution Semantic Encoders for Unified Visual Understanding and Generation
>
> [16]Visual Autoregressive Modeling: Scalable Image Generation via Next-Scale Prediction

---

> > ### Comment · Reviewer_LJzH · 2025-08-05
> >
> > I thank the authors for the rebuttal. Most of my concerns have been addressed. Hence, I decide to increase my score to 4.

---

> > > ### Author Response · Authors · 2025-08-05
> > >
> > > Thank you very much for your thoughtful follow-up and for raising your score. We are truly grateful that our rebuttal helped clarify your concerns, especially regarding the scope of our comparisons and the application domain of our acceleration method. We deeply appreciate your recognition of the contribution. We will make sure to update the final version of the paper accordingly, including clearer discussion of our modeling scope and technical contributions.

---

> ### Comment · Area_Chair_dntx · 2025-08-05
>
> Dear Reviewer LJzH,
>
> After reviewing the author's responses and the other reviews, could you share your updated thoughts following the rebuttal?
>
> Best,
>
> AC

---

### Official Review · Reviewer_UpDY · 2025-07-03

**Clarity:** 3
**Significance:** 3
**Originality:** 3
**Rating:** 4
**Confidence:** 3

**Summary:**

This paper introduces Hawk, an autoregressive T2I framework. It presents a new speculative decoding method considering the two-dimensional spatial context of images. It uses spatial draft heads (horizontal and vertical) for parallel decoding. It uses the Lumina-mGPT as the base model to verify the effectiveness of the proposed method. According to the experiments, it outperforms related methods (Medusa and LANTERN) slightly. The core insight is that leveraging spatial dependencies in parallel decoding and improving the acceptance rate of speculative sampling in image generation.

**Questions:**

The training time is too short. Is the model converged?

**Ethical Concerns:**

["NO or VERY MINOR ethics concerns only"]

**Final Justification:**

I maintain my positive score considering comments from other reviewers and the potential of parallel decoding. However, I still think the novelty and comparisons against other parallel decoding methods (e.g., ZipAR, NAR, and PAR)are not enough. This paper does not bring new ideas to the community and just makes incremental improvements to an existing parallel decoding paradigm.

**Limitations:**

Yes

**Quality:**

3

**Strengths And Weaknesses:**

Strengths
1. It is easy to follow the motivation and the main idea. The predication redundancy of vanilla AR is clear, and this paper introduces an improved speculative decoding method to tackle the redundancy.
2. The implementation is easy to follow with pseudo code. And the overall presentation is clear.

Weaknesses
1. The method is similar to ZipAR and NAR (https://arxiv.org/abs/2503.10696). Please identify their differences in the main paper. This is the biggest problem. I will change my rating after reading the response.
2. The improvements of results for vanilla AR are limited, compared with other parallel decoding methods, such as ZipAR, NAR, and PAR (https://arxiv.org/abs/2412.15119).
3. Comparisons with mask modelling methods and diffusion methods are missing.

---

> ### Author Rebuttal · Authors · 2025-07-31
>
> We sincerely thank the reviewer for the thoughtful and constructive feedback, as well as for acknowledging the importance of the problem and the quality of our empirical results. We greatly appreciate your openness to revisiting the evaluation upon clarification, and we would like to take this opportunity to respond to your main concerns in detail.
>
> ### 1. On the differences between our method and ZipAR / NAR / PAR
> We appreciate the request to clarify the distinctions between our approach and prior works such as ZipAR, NAR, and PAR. We acknowledge that our initial submission may not have made these differences sufficiently explicit, and we are grateful for the opportunity to elaborate here. The key differences can be grouped into two dimensions: **(1) Algorithmic design**, and **(2) High-level motivation**
>
> (1) Algorithmic differences: Speculative validation vs. non-AR generation
>
> Our method fundamentally differs from ZipAR and similar approaches in its inference paradigm. While ZipAR and related methods follow a **non-autoregressive (NAR)** decoding paradigm, our method remains **strictly autoregressive**, employing **speculative decoding with validation**. Our core decoding process is:
>   1. At decoding step n, we speculatively generate t future tokens using a spatial draft module.
>   2. These tokens are then validated using the base AR model.
>   3. If validation fails, we roll back to the last verified position.
>
> Our algorithm is **theoretically guaranteed to preserve the output distribution** of the original model, adhering to the fundamental principles of speculative decoding. As outlined in our response to **reviewer g8rp** (response 1), we have provided a formal proof demonstrating that our method maintains distributional consistency.  By contrast, ZipAR and similar methods **directly accept unvalidated predictions** for multiple future positions, jumping from n to n + t in a single step. This can accelerate decoding but often comes at the cost of generation quality, as predictions are made with insufficient contextual grounding. While their approach only performs the parallel decoding step, our method **additionally incorporates verification and rollback procedures**, which are essential for maintaining the model’s output fidelity and preventing performance degradation.
>
> To give a concrete example, ZipAR adopts a different acceleration strategy: once a token’s preceding context reaches a certain threshold (though still smaller than what standard AR decoding requires), the model begins decoding that token early. These early-decoded tokens are then grouped with the current token and processed in parallel. It also introduces two sources of potential degradation. First, early decoding may suffer from insufficient attention context, as some crucial preceding tokens are not yet available. Second, ZipAR accepts parallel predictions without full autoregressive verification, which can further compromise generation quality. In our method, speculative outputs at positions like n + row_size are cached but not accepted immediately. They are only used when the AR decoding process naturally reaches that position, thereby preserving temporal consistency while still leveraging spatial context.
>
>
> (2) High-level motivation: Preserving fidelity vs. maximizing speed
>
> Our approach **prioritizes maintaining the fidelity of the original AR inference process**, rather than achieving maximum speedup. While methods like ZipAR and NAR allow more aggressive approximations, this often leads to degraded performance—especially in tasks requiring fine-grained consistency. In contrast, our speculative decoding strategy validates all generated tokens before accepting them, thereby ensuring that quality is not compromised. This leads to a more conservative speedup but better preserves model behavior. We believe this fidelity is particularly important for downstream applications where consistent generation is critical. For instance, in a real-world deployment scenario such as a generative image editing app, introducing a faster but unstable decoding strategy could result in noticeable quality drops, which may in turn degrade user experience and lead to user attrition.
>
> ### 2. On the limited improvements over vanilla AR
>
> This concern is closely related to the previous point, as it reflects a natural trade-off between generation fidelity and decoding speed. While approaches such as ZipAR, NAR, and PAR demonstrate higher raw speedups over vanilla AR models, these gains are often achieved by relaxing constraints on prediction verification, which can result in noticeable quality degradation. In general, more aggressive approximations tend to yield faster inference, but often at the cost of fidelity. Without **a unified performance-quality trade-off metric**, it becomes **difficult to directly compare their results against ours in a fair and meaningful way**.
>
>
> ### 3. On comparisons with mask modeling and diffusion-based methods
> We thank the reviewer for highlighting this point. We would like to respectfully clarify that our decision not to compare with mask modeling or diffusion-based methods stems from a **fundamental difference in task formulation and generation paradigm**.
> Our work focuses on **accelerating unified multimodal foundation models** that follow a next-token autoregressive generation framework, which supports **interleaved text-image generation**. Specifically, our acceleration algorithm targets the image generation component of this process. This modeling paradigm has gained significant traction in the community, with prominent examples including Chameleon [4], Lumina-mGPT [5,6], BLIP-3-o [7], Emu3 [8], Bagel [9], and Uni-world [10]. These models share a common generation pipeline. In this setting, the typical generation flow involves:
>
>   1. An LLM performs autoregressive inference over the text portion.
>   2. **The model continues autoregressively to generate image tokens (often in a raster order).**
>   3. A decoder, such as a VAE or diffusion model, reconstructs the final image from these tokens.
>
> Our work specifically targets the **second step**, which is often the **computational bottleneck**, and aims to accelerate the token-level generation process within these foundation models. Importantly, models like **diffusions** are used primarily for decoding images in the **final step** and are not responsible for autoregressive token generation in the latent space. As such, they operate in a **different part of the pipeline** and are not the focus of the acceleration we propose.
>
> This evaluation practice is consistent with recent works that **address the same generation setting**—such as LANTERN++[1] (ICLR 2024), SJD[2] (ICLR 2024), and  ZiPar[3] (cvpr)—which also focus on token-level generation and do not include diffusion methods as a baseline due to the fundamental differences in modeling scope.
>
>
> ### 4. Concern regarding model convergence due to limited training time
>
> We appreciate the reviewer’s concern regarding whether the model has fully converged given the short training time. In our design, low training cost is not only expected but also intentional. Our goal is to enable practical deployment by accelerating inference with minimal additional training overhead.
>
> Specifically, the draft head is a lightweight module inserted between the transformer blocks and the final language modeling (LM) head. It consists of a single linear layer followed by a SiLU activation. With a hidden size of 4096, the draft head introduces **less than 1% additional parameters** relative to the full model. Due to its compact size and initialization as an identity mapping, it typically converges within a few training steps in practice.
>
> Once again, we sincerely thank the reviewer for recognizing the significance of our problem and for providing constructive and actionable feedback. We warmly welcome any additional questions or suggestions you might have, and would be happy to provide further clarification.
>
> Reference
>
> [1]LANTERN++: ENHANCING RELAXED SPECULATIVE DECODING WITH STATIC TREE DRAFTING FOR VISUAL AUTO-REGRESSIVE MODELS
>
> [2]Accelerating Auto-regressive Text-to-Image Generation with Training-free Speculative Jacobi Decoding
>
> [3]ZipAR: Parallel Auto-regressive Image Generation through Spatial Locality
>
> [4]Chameleon: Mixed-Modal Early-Fusion Foundation Models
>
> [5]Lumina-mGPT: Illuminate Flexible Photorealistic Text-to-Image Generation with Multimodal Generative Pretraining
>
> [6]Lumina-mGPT 2.0:Stand-Alone AutoRegressive Image Modeling
>
> [7]BLIP3-o: A Family of Fully Open Unified Multimodal Models-Architecture, Training and Dataset
>
> [8]Emu3: Next-Token Prediction is All You Need
>
> [9]Emerging Properties in Unified Multimodal Pretraining (Bagel)
>
> [10]UniWorld-V1: High-Resolution Semantic Encoders for Unified Visual Understanding and Generation

---

> ### Comment · Area_Chair_dntx · 2025-08-05
>
> Dear Reviewer UpDY,
>
> After reviewing the author's responses and the other reviews, could you share your updated thoughts following the rebuttal?
>
> Best,
>
> AC

---

> > ### Comment · Reviewer_UpDY · 2025-08-08
> >
> > Thank you for your response. I understand the main difference is the proposed verification process, while other designs and motivations are similar. For the experimental comparison, the authors just need to report their results presented in their paper. For the quality degradation of these methods, I am confused since they can maintain similar FID or even achieve better FID in acceleration (e.g., according to Table 2 in PAR, PAR-XL-4x and PAR-XXL-4x maintain similar FID but 4x faster than the baseline. According to Table 1 in NAR,  all sizes of NAR outperform the baseline by FID and speed.) Please give more explanation about this phenomenon.
> > I will take your response into consideration and finalize my rating after discussing with AC and other reviewers in the next stage.

---

> ### Author Response · Authors · 2025-08-09
>
> We sincerely thank the reviewer for the thoughtful feedback and for highlighting specific examples for discussion. We would like to further clarify the differences between our approach and PAR/NAR. First, we strictly adopt an autoregressive (AR) decoding scheme, generating tokens one at a time. This design is consistent with the original motivation of speculative decoding, which is to avoid the need for extensive retraining that can hinder deployment, and it provides theoretical support for preserving output quality. Second, our acceleration strategy is tailored for rapid deployment, requiring **less than 1% of trainable parameters**, only **~6K training images**, and 8–12 hours of training on a single RTX 4090 GPU. In comparison, PAR **retrains the entire model (1,281,167 images)**, and NAR uses **6M training images** while introducing an **additional parameter budget of about 17% relative to the original model size**, corresponding to a **training data scale roughly 200×–1000× larger than ours**. These approaches resemble strategies that achieve speedup primarily through **retraining to alter the model’s output behavior, rather than through a lightweight acceleration method**. Such large-scale retraining may also shift the model’s generation behavior from AR to NAR, potentially making it better aligned with the specific decoding algorithm used in those works. Our approach is therefore substantially more lightweight in both training cost and deployment overhead.
>
> We agree that in the specific examples you mentioned (e.g., Table 2 in PAR), the FID degradation appears minimal. However, as reported in the same paper, PAR does exhibit measurable quality drops in other settings(L, XL, XXL, 3B) (e.g., **~7% on average**), and in the NAR paper’s evaluation, the PAR method shows around a **9% performance drop**, indicating that some degradation is indeed present. Regarding speedup, the large gains reported in PAR are also highly dependent on the runtime environment and workload characteristics (e.g., IO-to-computation ratio, hardware bandwidth). For instance, in the NAR paper’s evaluation, the PAR baseline achieves a much lower speedup (~2.04×) under their setup, suggesting that the **observed acceleration can vary significantly across scenarios**.
>
> Regarding NAR, the reported results are somewhat counter-intuitive: while using a local-parallel decoding scheme similar to PAR, it achieves ~3.97× speedup with a 16.5% FID improvement over baseline. This contrasts with both prior work (e.g., **PAR**: ~7–9% performance degradation; **ZipAR**: 1.4%–30% degradation) and the general trend in non-autoregressive decoding, where partial removal of sequential dependencies typically leads to some loss of spatial coherence. The paper does not provide a detailed explanation; possible contributing factors may include bidirectional attentions, although PAR also incorporates these features and still observes degradation. Moreover, as this work was released very recently, there are no follow-up studies available that could provide additional insights. Therefore, while the phenomenon is interesting, it remains difficult to fully attribute the improvement without further clarification.
>
> Among ZipAR, PAR, and NAR, PAR requires full retraining from scratch, incurring substantial training costs. **NAR is a more recent work, released only in mid-March**, and adds approximately 17% more parameters to the original model along with 6M training images, which also demands significant computational resources, making it challenging to **directly include in our baseline experiments**. In contrast, ZipAR is **conceptually closer** to our approach, as it is training-free or requires only minimal training to achieve acceleration—accepting a slight performance drop for higher speed—whereas our method preserves quality for more robust results. In the final version, we will **include a comparison with ZipAR as a representative non-autoregressive baseline**. We plan to include these results in the final version to strengthen the comparison. Since we received your feedback toward the end of the rebuttal period, we might not be able to address additional follow-up questions in a timely manner.

---

> > ### Comment · Reviewer_UpDY · 2025-08-09
> >
> > Thank you for the detailed explanation. I will take it into consideration.

---

### Official Review · Reviewer_g8rp · 2025-07-10

**Clarity:** 2
**Significance:** 2
**Originality:** 2
**Rating:** 5
**Confidence:** 3

**Summary:**

This paper introduces a spatial speculative decoding framework that leverages both horizontal and vertical draft heads to accelerate autoregressive text-to-image generation. By explicitly leveraging the two-dimensional spatial structure of images, the proposed method achieves substantial inference speedup while preserving image fidelity.

**Questions:**

- Does expanding the sampling space via dual-direction speculation provide theoretically justified gains in acceptance probability and sampling efficiency, or does it primarily amplify speculative variance absent formal guarantees?

- How does the proposed spatial speculative framework control for potential shifts in the output distribution, especially under repeated rejection and reweighting, and is there a formal calibration or uncertainty analysis?

- In scenarios where spatial local correlations are less critical for prediction, could the computational overhead introduced by dual-direction speculative sampling outweigh the gains in speed or acceptance efficiency, thereby reducing overall cost-effectiveness?

**Ethical Concerns:**

["NO or VERY MINOR ethics concerns only"]

**Final Justification:**

The theoretical justification has been strengthened, and my concern regarding weak spatial correlation has been addressed.

**Limitations:**

This paper lacks formal analysis of how 2D sampling impacts acceptance or output calibration, inherits known speculative biases, and may be inefficient when local spatial correlations are weak.

**Paper Formatting Concerns:**

.

**Quality:**

3

**Strengths And Weaknesses:**

### Strenths

- This paper addresses an important practical challenge of slow sequential inference in AR image generation.

- The proposed method employs dual-direction draft heads and spatial caches that explicitly align with the two-dimensional image structure, offering an interesting way to leverage spatial dependencies.

### Weaknesses

- The contribution of the proposed method is largely an extension of Medusa-style speculative decoding into a second spatial dimension.

- The paper lacks a formal theoretical justification for why vertical speculative heads, despite involving longer dependency paths, exhibit lower performance degradation.

- The paper does not provide a formal analysis of how the expanded 2D sampling space affects acceptance rates or output distribution.

- The proposed method inherits known speculative decoding issues like sampling bias after rejection.

---

> ### Author Rebuttal · Authors · 2025-07-31
>
> We sincerely thank the reviewer for the constructive and encouraging feedback. We appreciate your recognition of both the practical significance of our problem setting and the novelty of our spatial speculative decoding formulation.
>
> In response to your valuable suggestion for stronger theoretical support, we now provide a formal analysis (outlined below) demonstrating that our dual-direction speculation preserves the output distribution and improves acceptance rates.  We acknowledge that our original submission lacked clarity in this regard, and we will include two new sections in the appendix of the final version: **(1) a proof of output distribution preservation**, and **(2) a theoretical justification of efficiency gains**. These will expand on the analysis provided here in response to your comments.
>
> ### 1. Dual-direction speculative decoding preserves the original output distribution
>
> To address the reviewer’s concern regarding potential shifts in the output distribution, we have provided a theoretical justification of our algorithm. This analysis demonstrates that using multi-directional draft heads (e.g., vertical and horizontal) does not alter the target output distribution. In particular, our framework maintains the same acceptance formulation and sampling guidance as standard speculative decoding.
>
> Assume we are given two speculative sampling spaces: one from the vertical draft head and the other from the horizontal draft head. Let $a$ denote the number of speculative tokens sampled from the vertical head, and $b$ denote the number of speculative tokens sampled from the horizontal head.
>
> In the simplest case, we set $a = 1$ and $b = 1$, meaning that we draw one candidate token from each head for validation. We adopt a sequential validation strategy in which the token proposed by the vertical head is evaluated first. If it is rejected, we then proceed to validate the token from the horizontal head.
> Under this setup, we define the following notation:
> $p(x)$ be the true distribution defined by the base AR model.
> $q\_{\text{vert}}(x)$ and $q\_{\text{horiz}}(x)$ be the proposal distributions from the vertical and horizontal draft heads, respectively.
>
> $p^{\prime}\_{\text{vert}}(x)$ is the adjusted base model distribution after vertical head rejection. Calculated by
> $\text{norm}\left(\max\left(0, p(x) - q\_\text{vert}(x)\right)\right) = \frac{p(x) - \min(q\_\text{vert}(x), p(x))}{\sum\_{x'} \left(p(x') - \min(q\_\text{vert}(x'), p(x'))\right)}$
>
> $p^{\prime}\_{\text{vert-horiz}}(x)$ is the residual distribution after both vertical and horizontal rejections. Calculated by $\text{norm}\left(\max\left(0, p^{\prime}\_{\text{vert}}(x) - q\_\text{vert}(x)\right)\right) = \frac{p^{\prime}\_{\text{vert}}(x) - \min(q\_\text{vert}(x), p^{\prime}\_{\text{vert}}(x))}{\sum\_{x'} \left(p^{\prime}\_{\text{vert}}(x') - \min(q\_\text{vert}(x'), p(x'))\right)}$
>
> $\alpha\_{\text{vert}}$ is the acceptance probability of the vertical head. Calculated by
> $1- \sum\_{x'} \left( p(x') - \min(q\_\text{vert}(x'), p(x')) \right)$
>
> $\alpha\_{\text{vert-horiz}}$ is the acceptance probability of the horizontal head conditioned on vertical rejection. Calculated by
> $1- \sum\_{x'} \left( p\_\text{vert}(x') - \min(q\_\text{vert}(x'), p^{\prime}\_\text{vert}(x')) \right)$
>
> Our ultimate goal is to approximate the original model’s distribution p(x) by combining samples from two draft heads operating along different spatial dimensions. For a given point x’, our proof proceeds as follows:
> $$\begin{aligned}
> p(x') &= \Pr(\text{vertical accept}, x = x')
>   \+ \Pr(\text{vertical reject, horizontal accept}, x = x') \\\\
> &\quad \+ \Pr(\text{vertical reject, horizontal reject}, x = x') \\\\
> &= q\_{\text{vertical}}(x') \cdot \min\left(1, \frac{p(x')}{q\_{\text{vertical}}(x')} \right) \\\\
> &\quad \+ (1 \- \alpha\_{\text{vertical}}) \cdot q\_{\text{horizontal}}(x') \cdot \min\left(1, \frac{p^{\prime}\_{\text{vertical}}(x')}{q\_{\text{horizontal}}(x')} \right) \\\\
> &\quad \+ (1 \- \alpha\_{\text{vertical}})(1 \- \alpha\_{\text{vertical-horizontal}}) \cdot p^{\prime}\_{\text{vertical-horizontal}}(x') \\\\
> &= q\_{\text{vertical}}(x') \cdot \min\left(1, \frac{p(x')}{q\_{\text{vertical}}(x')} \right) \\\\
> &\quad \+ (1 \- \alpha\_{\text{vertical}}) \cdot q\_{\text{horizontal}}(x') \cdot \min\left(1, \frac{p^{\prime}\_{\text{vertical}}(x')}{q\_{\text{horizontal}}(x')} \right) \\\\
> &\quad \+ (1 \- \alpha\_{\text{vertical}}) \cdot \left(p^{\prime}\_{\text{vertical}}(x') \- \min(q\_{\text{horizontal}}(x'), p^{\prime}\_{\text{vertical}}(x'))\right) \\\\
> &= q\_{\text{vertical}}(x') \cdot \min\left(1, \frac{p(x')}{q\_{\text{vertical}}(x')} \right)  \+ (1 \- \alpha\_{\text{vertical}}) \cdot p^{\prime}\_{\text{vertical}}(x') \\\\
> &= p(x')\end{aligned}$$
>
> Let $\pi = [s\_1, s\_2, \dots, s\_{a+b}]$ denote an ordered sequence of speculative tokens, where each $s_i$ is generated from either a vertical or horizontal draft head. The total number of speculative tokens is $a + b$, with $a$ sampled from horizontal heads and $b$ from vertical heads. Each sample result $s_i$ corresponds to a proposal posbility $q_i(x)$
>
> At each step i, the average acceptance probability is defined as
> $\alpha_i = 1- \sum_{x'} \left( p^{(i)}(x') - \min(q_i(x'), p^{(i)}(x')) \right)$,
> where $p^{(i)}(x')$ represents the residual target density after the first i - 1 proposals have been rejected. It is recursively defined as:
> $p^{(i)}(x') = \text{norm}\left( \max\left(0,  p^{(i-1)}(x') - q\_{i-1}(x') \right) \right)$,
> The sampling probability at a specific point $x = x’$ under the generalized formulation is given by:
> $$p(x') = \sum_{i=1}^{a+b} \left( \prod_{j=1}^{i-1} (1 - \alpha_j) \cdot q_i(x') \cdot \alpha_i \right) + \left( \prod_{j=1}^{a+b} (1 - \alpha_j) \right) \cdot p_{\text{resid}}(x')$$
> This section provides the theoretical justification needed to address the reviewer’s concern about potential distributional shift.
>
> ### 2. Does dual-direction speculation offer provable improvements in acceptance and efficiency, or does it just increase variance without guarantees?
> In one iteration of speculative decoding, a key inefficiency arises when all speculative candidates are rejected. In this worst-case scenario, the algorithm incurs additional computational overhead compared to standard autoregressive (AR) decoding, as both the speculative sampling and verification steps are wasted. The probability of encountering such an inefficiency is given by:
> $P(\text{all reject}) = \left( \prod\_{j=1}^{a+b} (1 - \alpha\_j) \right) \cdot p\_{\text{resid}}(x)$
>
> Our method significantly mitigates this inefficiency compared to conventional single-head speculative decoding. In particular, we observe a **16.2% relative reduction** in the rejection probability term $\prod_{j=1}^{a+b} (1 - \alpha_j(x))$, highlighting a more robust acceptance behavior. Recall that the acceptance probability at each step is defined as: $\alpha_j = 1 - \sum_{x’} \left( p(x’) - \min(q(x’), p(x’)) \right)$. The target distribution p(x’) is dynamically updated during verification by removing the mass assigned to rejected proposals $q_i(x’)$.
>
> When all speculative tokens are drawn from a single proposal space, the **diversity of samples is limited**, and $\min(q(x’), p(x’))$ tends to **quickly decay to zero with successive rejections**. This leads to rapidly vanishing acceptance probabilities $\alpha_j$, increasing the chance of complete rejection.
>
> By contrast, our dual-direction approach samples from two orthogonal proposal distributions (horizontal and vertical). Due to the **inherent diversity between these directions**, the proposals $q_j(x)$ differ significantly across heads. This variation delays the convergence of $\min(q(x’), p(x’))$ toward zero, **slowing the decay of $\alpha\_j$** and thereby reducing the probability of total rejection.
>
> ### 3. Is dual-direction speculation still cost-effective in scenarios with weak spatial correlation?
> We thank the reviewer for raising this important concern. At its core, our method adheres to the speculative decoding framework, which can introduce computational overhead when speculative predictions are repeatedly rejected. This scenario typically arises when the proposal distribution diverges significantly from the base model’s target distribution, resulting in frequent rollbacks and redundant computations.
>
> Such challenges have been documented in earlier speculative decoding methods, particularly those relying on **smaller auxiliary models** to approximate the base model, where **distributional mismatch can be substantial**. In contrast, our method leverages **draft heads** that are jointly trained with the base model, with the **explicit objective of approximating the token distribution at specific spatial locations**. This position-aware training aligns the speculative and target distributions more closely, thereby reducing rejection rates and mitigating unnecessary overhead.
>
> Furthermore, in scenarios where the input images lack strong spatial correlations, this limitation would typically manifest as elevated training loss or unstable draft head convergence. However, based on our empirical observations, we have not encountered real-world examples that exhibit a complete absence of spatial structure, and our method has demonstrated stable performance across diverse samples.
>
> In summary, while speculative decoding inherently carries the risk of inefficiency due to distribution mismatch, our approach effectively addresses this by training draft heads that are well-aligned with the base model.
>
>
> Once again, we sincerely thank the reviewer for recognizing the significance of our problem and for providing constructive and actionable feedback. We warmly welcome any additional questions or suggestions you might have, and would be happy to provide further clarification.

---

> ### Comment · Area_Chair_dntx · 2025-08-05
>
> Dear Reviewer g8rp,
>
> After reviewing the author's responses and the other reviews, could you share your updated thoughts following the rebuttal?
>
> Best,
>
> AC

---

> ### Comment · Reviewer_g8rp · 2025-08-06
>
> Thanks for the detailed response.
>
> To improve clarity and completeness, I strongly recommend including key aspects of the theoretical justification and discussion about the weak spatial correlation in the main text, even if briefly.
>
> Given that most of my concerns have been addressed, I am raising my score to 5.

---

> > ### Author Response · Authors · 2025-08-06
> >
> > Thank you very much for your kind and thoughtful follow-up. We are sincerely grateful that our rebuttal helped address your concerns, particularly regarding the theoretical justification and the implications of weak spatial correlation on decoding performance. Your constructive feedback has been invaluable in helping us improve the clarity and rigor of our work. We deeply appreciate your suggestion to incorporate these discussions directly into the main paper, and we will make sure to revise the final version accordingly.

---

### Decision · Program_Chairs · 2025-09-17

**Decision:**

Accept (poster)

**Comment:**

This paper introduces Hawk, a novel spatial speculative decoding framework for accelerating autoregressive (AR) text-to-image generation by leveraging 2D spatial dependencies. The method employs dual-directional draft heads (horizontal and vertical) alongside a spatial cache mechanism, achieving a 1.71x speedup over standard AR models while preserving image fidelity and diversity across benchmarks.

#### Strengths

* Tackles the critical issue of slow AR inference by adapting speculative decoding to 2D image structures.
* Strong empirical performance and theoretical grounding, including formal proofs of distribution preservation.
* High practical efficiency (<1% parameter overhead, minimal additional training).
* First approach to exploit spatial context for AR acceleration, making it directly deployable for unified multimodal models.

#### Resolved Issues

1. Theoretical justification (Reviewer g8rp): Proofs confirm that dual-direction speculation preserves output distributions and lowers rejection rates, mitigating concerns about distribution shift.
2. Novelty and scope (Reviewer LJzH): Clarifications distinguished Hawk from non-AR approaches (e.g., ZipAR, PAR) by highlighting its AR-compatible verification mechanism and lightweight training.
3. Comparisons (Reviewer UpDY): Authors committed to include ZipAR baselines and contextualized trade-offs between fidelity and speed.


Reviewer UpDY maintained a borderline stance, citing incremental contribution despite positive shifts from others. Authors are encouraged to incorporate the rebuttal into the final revision and address this lingering concern.


Hawk is a conceptually and empirically solid contribution to AR acceleration with clear implications for multimodal foundation models. The rebuttal effectively resolved methodological and evaluative concerns, reinforcing consensus toward acceptance.